# Filter & Align: Leveraging Human Knowledge to Curate Image-Text Data

## Abstract

The increasing availability of image-text pairs has largely fueled the rapid advancement in vision-language foundation models. However, the vast scale of these datasets inevitably introduces significant variability in data quality, which can adversely affect the model performance. This highlights the critical role of data filtering, not only to enhance training efficiency but also to improve overall data quality. Existing methods typically rely on metrics such as CLIP Score and BLIP Score, which are derived from pre-trained models. However, these models are often trained on uncurated, noisy datasets, which can perpetuate errors and misalignments in the filtered dataset. We present a novel algorithm that incorporates human knowledge on image-text alignment to guide filtering vast corpus of web-crawled image-text datasets into a compact and high-quality form. To systemically capture human preferences on image-text alignments, we collect a diverse image-text dataset where each image is associated with multiple captions from various sources, and establish a comprehensive set of both subjective and objective criteria for critically guiding the alignment assessment from labelers. Additionally, we train a reward model on these human-preference annotations to internalize the nuanced human understanding of image-text alignment. The resulting reward model thus can act as a human-like referee to filter image-text pairs. Extensive experiments demonstrate that we can maintain, sometimes even improve, model performance while compressing the image-text datasets up to ∼90%. An impressive example is that, by aggressively reducing the total training sample from 130M to only 15.5M, our BLIP-B/16 models consistently show an average improvement of 2.9% on retrieval tasks and 11.5% on captioning tasks compared to full-size-dataset counterparts.

## 1 Introduction

The rapid progress in vision-language foundation models (Alayrac et al., 2022; Changpinyo et al., 2021; Jia et al., 2021; Sharma et al., 2018; Radford et al., 2021; Pham et al., 2023) has been largely driven by the growing availability of image-text pairs, experiencing a massive escalation from datasets comprising a few million samples, such as COCO (Lin et al., 2014), CC3M (Sharma et al., 2018) and CC12M (Changpinyo et al., 2021), to those encompassing billions, exemplified by YFCC-100M (Thomee et al., 2016), LAION-400M (Schuhmann et al., 2021), and LAION-5B (Schuhmann et al., 2022). Typically, to facilitate the large-scale collection of such data, web crawling is applied with simple filtering mechanisms in place. While this collection pipeline ensures a rich diversity of data, it inadvertently introduces significant variability in data quality, presenting new challenges for learning at scale.

A popular strategy for mitigating this challenge is data filtering. Recent literature presents various methodologies where models are designed to assess the alignments between images and their corresponding textual descriptions, selectively retaining only those pairs that meet predefined standards (Wang et al., 2023; Xu et al., 2023a). However, a fundamental concern associated with this line of work is that the filtering models are typically built upon the original, uncurated datasets, which are inherently noisy. This noise can propagate biases and misalignments into filtered datasets, potentially impinging upon the top-line performance for subsequent models trained on such data. Moreover, the intrinsic cognitive discrepancies between human and machine perception, as studied in recent works (Otani et al., 2023; Lee et al., 2023; Ruiz et al., 2023),

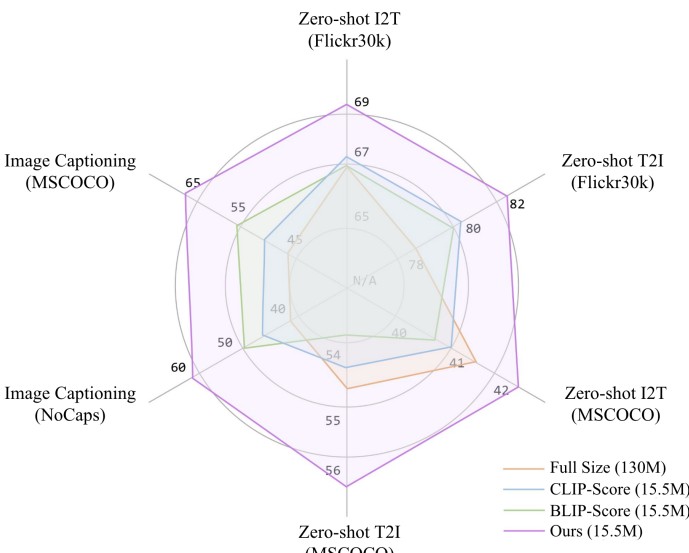

Figure 1: Our method outperforms full-size training dataset on various downstream tasks with BLIP-B/16. This training set consists of CC3M, CC12M, and a subset of LAION-400M. We reduce the training sample size **from 130M to 15.5M (*i.e.* ~9×smaller)**.

suggest that machine-based assessments alone may not sufficiently encapsulate the quality standards set by human judgment.

Intriguingly, recent advancements in language models (Bahdanau et al., 2017; Kreutzer et al., 2018; Nakano et al., 2021; Zhou & Xu, 2020; Ouyang et al., 2022) have demonstrated that Reinforcement Learning from Human Feedback (RLHF) (Christiano et al., 2017; Stiennon et al., 2020) can effectively incorporate human preferences as a reward signal, markedly aligning the model with human intentions. A key component here is the reward model, which is trained to approximate the often complex and nuanced human evaluations of desired behavior. By integrating human feedback directly into the training loop, models can develop a more profound understanding of complex domains of human knowledge. Inspired by these advancements, this paper seeks a human-centric approach to the curation of image-text pairings, aiming to improve data quality through a filtration process intrinsically attuned to the subtleties of human cognition and evaluative criteria.

An overview of our data filtering pipeline, which is grounded in human knowledge, is illustrated in Figure 2. Specifically, our first step is to collect a dataset with 1,000 images, each paired with a variety of captions. Then, human preferences are collected by ranking the alignment between different captions to their corresponding images, applying a set of criteria that captures both objective aspects, such as accuracy and completeness, and subjective aspects, such as vividness and contextual relevance. In the final stage, we train a reward model on the annotated dataset, aiming to predict the human preference for captions. The resulting reward model is expected to function as a human-like referee that can transfer human knowledge to identify and filter the misaligned and low-quality image-text pairs for enhancing the overall dataset quality.

Extensive experiments are provided to demonstrate that we can successfully compress large-scale and noisy image-text datasets into compact and well-aligned forms. A compelling illustration of our method's capability is detailed in Figure 1, where we demonstrate the significant compression of a composite dataset—comprising elements from CC3M (Sharma et al., 2018), CC12M (Changpinyo et al., 2021), and a subset of LAION-400M (Schuhmann et al., 2021)—from an initial count of 130M samples down to just 15.5M. Remarkably, when training BLIP-B/16 model, our method achieves an average improvement of 2.85% on retrieval tasks and 11.5% on captioning tasks, compared to the full-sized dataset. Additionally, our method is notably higher than the -0.07% and 2.1% improvements with the BLIP Score and -0.3% and 9.0% improvement with the CLIP Score. These findings underscore the potential of our human-centric approach to refine and

enhance the utility of image-text datasets significantly, paving the way for more efficient and effective model training in the vision-language domain.

## 2 Related Works

### 2.1 Learning from human knowledge.

The integration of human knowledge is becoming progressively instrumental in aligning model behavior with human intent (Bahdanau et al., 2017; Kreutzer et al., 2018; Nakano et al., 2021; Zhou & Xu, 2020; Xu et al., 2023b; Lee et al., 2023; Wu et al., 2023; Zhang et al., 2023). The key part is to train a reward model (MacGlashan et al., 2017) to directly capture human preferences regarding the outputs generated by the model. Recent works propose to utilize reinforcement learning (Schulman et al., 2017) to finetune the language models (Ouyang et al., 2022) and diffusion models (Xu et al., 2023b; Lee et al., 2023; Zhang et al., 2023) with the signal of reward model. However, few studies focus on utilizing human knowledge to directly improve dataset quality. This work ventures into this topic, pioneering the exploration of integrating human knowledge with image-text data to relieve the visual and textual misalignment in large-scale image-text datasets.

### 2.2 Vision-language data.

Data is of central importance to the success of vision-language pre-training at scale (Jia et al., 2021; Alayrac et al., 2022; Radford et al., 2021). Early efforts in dataset collection often employ simple but vague cleaning strategies (Sharma et al., 2018; Changpinyo et al., 2021; Thomee et al., 2016). Recently it has shifted towards a more meticulous examination of visual-textual congruence, aiming to filter out misaligned image-text pairs more effectively. For instance, LAION-400M (Schuhmann et al., 2021) utilized similarity scores of pre-trained CLIP model (Radford et al., 2021) to filter low-quality image-text pairs. Recently, a series of works have investigated based on the similarity scores of CLIP and BLIP models. CiT (Xu et al., 2023a) dynamically curates the data during training a CLIP model on the fly. Since the proposed dynamic compress method is tied to the adjusting distribution of the CLIP model, retraining is necessary for each new dataset. TL;DR (Wang et al., 2023) initially employs the BLIP model to screen out low-quality image-text pairs coarsely. Subsequently, it learns a codebook to cluster and refine the filtering process. These data-filtering models are typically built upon noisy datasets, therefore potentially being less effective in filtering data.

## 3 Method

An overview of our human-knowledge-based image-text filtering pipeline is illustrated in Figure 2. We first generate a set of diverse captions for a given set of images, subsequently soliciting human assessments to determine the degree of alignment between each image and its associated captions. This phase involves a comprehensive scoring process by labelers to evaluate the quality of image-caption alignment, as detailed in Section 3.1. Next, we train a reward model to emulate human evaluative feedback on image-text pairs, as described in Section 3.2. The resulting reward model is expected to function as a human-like referee to improve image-text datasets (see Section 3.3).

### 3.1 Human Knowledge Collection

We construct a systematic pipeline to build an image-text dataset enriched with human feedback, as delineated in Step 1 of Figure 2. This pipeline contains two integral components: the collection of image-caption data and the subsequent annotation of this data with human feedback provided by expert labelers. This built image-caption dataset is expected to be both well-aligned and diverse: on an individual level of image-text pair, the caption contains rich and useful visual information accurately reflecting the image's content; on a population level of dataset, the distribution of captions is diverse preventing biases from a single source.

**Image-Captions Collection.** We utilize MSCOCO (Lin et al., 2014) as the basis to curate our new dataset, termed *COCO-HF*. Since MSCOCO contains images of 80 categories, we randomly sample 1,000

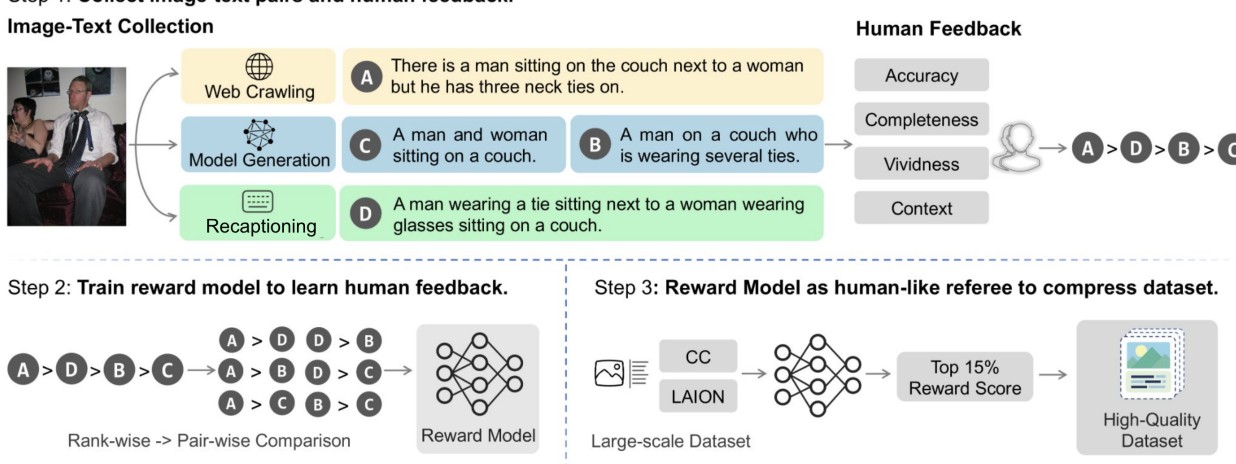

Figure 2: A diagram illustrating the three steps of our method. We first curate an image-text dataset to collect human knowledge on alignment in Step 1. Then we train a reward model to predict human preference in Step 2. The reward model functions as a human-like referee to filter misaligned image-text pairs in Step 3.

images from the MSCOCO dataset, evenly distributed across the categories. To enrich the diversity of captions, we explore the following strategies:

- **MSCOCO Sampling.** We sample the text descriptions in the MSCOCO dataset. In the MSCOCO dataset, each image is associated with five distinct human-write text descriptions.

- **Model Generation.** We employ various generative models, such as BLIP (Li et al., 2022), BLIP-2 (Li et al., 2023), and InstructBLIP (Dai et al., 2023) to generate captions using images as inputs and prompts optionally.

- **Human Recaption.** We engage human annotators and task them with rewriting captions based on content depicted in the corresponding observed images.

With these three steps, we collect a dataset with 1,000 candidate images, each accompanied by 8 to 10 captions.

**Human-Preference Annotations.** We recruit well-trained human labelers to evaluate the alignment between generated captions and image content. To ensure annotation consistency, we instituted four precise guidelines that outline the criteria for alignment:

- **Accuracy.** The caption should accurately reflect the content of the corresponding image. The inaccurate captions include descriptions conflicted with the content of the image, fabrication not relying on corresponding images, *etc.*

- **Completeness.** The caption should contain the main visual objects in the image as completely as possible. It guarantees the caption acknowledges an abstract view of the whole image.

- **Vividness.** The caption should describe the details of the mentioned visual objects. The details here refer to the number, appearance, action, and status of the visual objects. The vivid captions provide individual status and relationships between different visual objects, which is a kind of comprehensive understanding of the image.

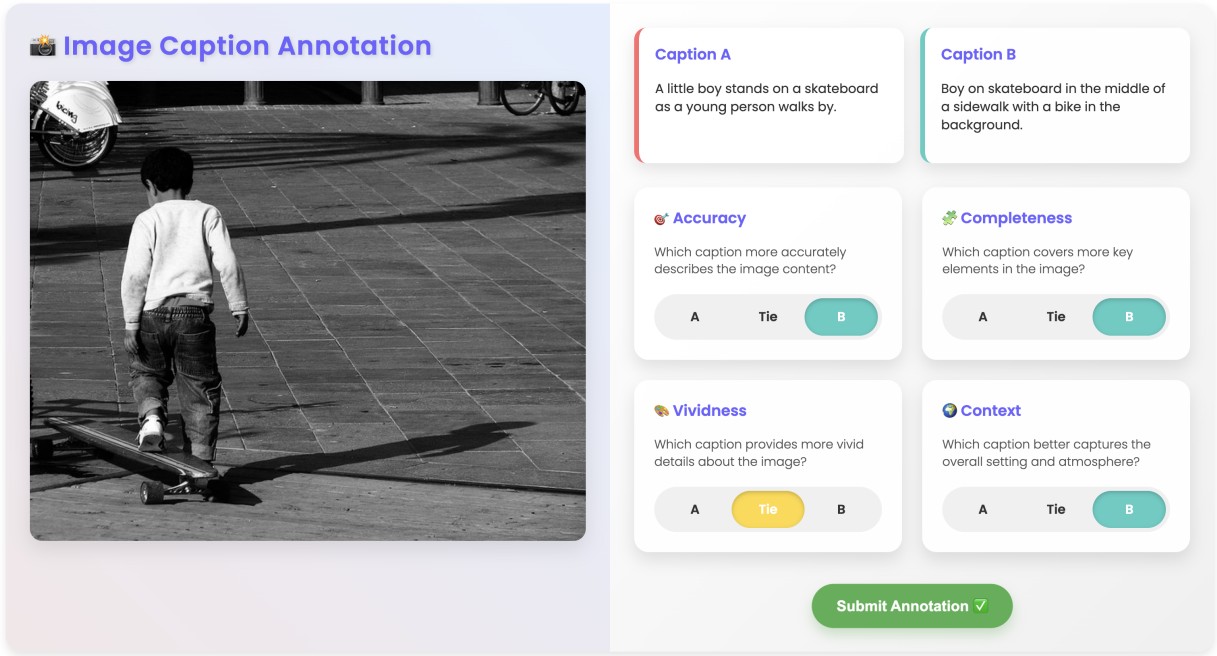

Figure 3: Annotation interface for image caption evaluation. Annotators compare two captions (A and B) for a given image across four criteria (Accuracy, Completeness, Vividness, and Context) respectively.

- **Context.** The caption should mention the context of the image, which is easily ignored. The context includes the background where the image occurs and the atmosphere that the image conveys or implies. It contains complete observation and human understanding of the image.

These criteria collectively contribute to a multi-faceted annotation framework that captures both the objective denotations and the subjective connotations of image-text pair.

## 3.2 Reward Model Training

Inspired by InstructGPT (Ouyang et al., 2022), we propose to train a reward model to learn the human preference knowledge on this newly developed COCO-HF dataset. This reward model is designed to capture the subtleties of human preferences, thereby serving as an automatic, yet human-like, arbiter for aligning image-text correspondences (seen in Figure 2 Step 2).

Specifically, we transform the preference annotations as rankings and formulate the training of the reward model as a pairwise ranking problem. For each given image $I$ in the COCO-HF dataset, we have $m \in [8, 10]$ textual captions ranked by human labelers, which are denoted as $x_1, x_2, ..., x_m$. If $x_i$ is better than $x_j$, we organize $(I, x_i, x_j)$ as a comparison pair. This produces at most $C_m^2$ comparison pairs for each image. Then, we follow the Bradley-Terry model (Bradley & Terry., 1952; Ouyang et al., 2022) of preferences to define the pair-wise loss function as:

$$\text{loss}(\theta) = -\mathbb{E}_{(I,x_i,x_j)\sim\mathcal{D}_H} \left[ \log(\sigma(f_\theta(I, x_i) - f_\theta(I, x_j))) \right],$$

where $f_\theta(I, x)$ is a scalar value of reward model $f$ parameterized by $\theta$ for image $I$ and caption $x$, $\sigma$ is the sigmoid function, and $\mathcal{D}_H$ denotes COCO-HF dataset.

**Implementation.** Following (Xu et al., 2023b), our reward model $f_\theta$ contains BLIP (Li et al., 2022) as the backbone and score mapping layer based on MLPs. The backbone produces a multi-modal embedding

of image and text features, and the score mapping layer maps the multi-modal embedding to a scalar as the reward score. When training this reward model, we freeze the parameters of the backbone BLIP and take the MLP part as trainable. The hyperparameter setup and training details of reward model training are provided in the supplementary material.

### 3.3 Dataset Compression

In order to compress large-scale datasets into compact and well-aligned ones, we utilize the reward model as a human-like referee to filter misaligned image-text pairs (Step 3 in Figure 2).

Let $\mathcal{D} = \{(I^n, x^n)\}_{n=1}^N$ be the original image-text dataset. We utilize the reward model $f_\theta$ to evaluate the image-text alignment of each image-text pair $(I^n, x^n)$ from the dataset $\mathcal{D}$. The reward score $r_i$ is formulated as $r^n = f_\theta(I^n, x^n)$. Then, we obtain the set of reward scores $R_\mathcal{D}$ of the whole dataset $\mathcal{D}$, which can be expressed as $R_\mathcal{D} = \{r^1, r^2, ..., r^N\}$. In order to compress the original dataset $\mathcal{D}$ to a compact and well-aligned dataset $\hat{\mathcal{D}}$ with $k\%$ original amount, we consist $\hat{\mathcal{D}}$ of image-text pairs with top $k\%$ reward score in $R_\mathcal{D}$. We train the vision-language models with compressed dataset $\hat{\mathcal{D}}$ and evaluate on various downstream tasks.

## 4 Experiment

In this section, we demonstrate the effectiveness of the proposed method by training vision-language models on compressed datasets and evaluating their performance across various downstream tasks.

### 4.1 Training and Evaluation

**Datasets.** To evaluate the effectiveness of our method, we conduct experiments on a diverse range of multimodal datasets. These include large-scale web-crawled datasets such as Conceptual Captions (Sharma et al., 2018), Conceptual 12M (Changpinyo et al., 2021), LAION-400M (Schuhmann et al., 2021), and SBU captions (Ordonez et al., 2011), which collectively comprise over 415 million image-text pairs. We also incorporate smaller, human-annotated datasets like COCO (Lin et al., 2014) and Visual Genome (Krishna et al., 2017). to provide a comprehensive evaluation across different data types and scales.

**Baselines.** We compare our method against several baseline approaches to assess its relative performance:

- **Full Size.** We use the entire dataset without any filtering.

- **Random.** We compress the dataset by randomly selecting 50% of the entire dataset.

- **CLIP Score and BLIP Score.** We experiment with CLIP Score and BLIP Score and leverage the cosine similarity between image and text embeddings as the filtering metric. Specifically, we select the image-text pairs that rank within the top 50%. The CLIP model and BLIP model are both equipped with ViT-L/14 image encoder. The CLIP model is pre-trained on the DataComp-1B dataset, while the BLIP model is trained on a diverse corpus of 129 million image-text pairs, which include Conceptual Captions, Conceptual 12M, LAION-400M, SBU captions, COCO, and Visual Genome.

**Training and Evaluation Setup.** We employ the BLIP (Li et al., 2022) of ViT-B/16 (Dosovitskiy et al., 2020) architecture and we train the models with the AdamW optimizer (Kingma & Ba, 2015). The batch size is set as 600 for CC3M dataset and 2,880 for CC12M dataset. The training epoch is 40, the learning rate is set to $3 \times 10^{-3}$, and weight decay is set to 0.1. We consider three downstream evaluation tasks for the trained BLIP models: image classification, image-text retrieval, and image captioning tasks. Importantly, to avoid possible bias introduced in fine-tuning, we evaluate all these tasks in a zero-shot manner.

| Method | Image-Text Retrieval | | | | Image Classification | | Image Captioning | | | |
| | Flickr30K | | MSCOCO | | ImNet-1K | ImNet-A | MSCOCO | | NoCaps | |
| | TR@1 | IR@1 | TR@1 | IR@1 | Acc. | Acc. | BLEU@4 | CIDEr | CIDEr | SPICE |
|---|---|---|---|---|---|---|---|---|---|---|
| **CC3M (2.82M → 1.41M)** | | | | | | | | | | |
| *Full Size* | 66.0 | 51.9 | 39.9 | 28.8 | 29.0 | 10.0 | 7.3 | 9.3 | 34.1 | 6.9 |
| Random | 57.7 | 44.4 | 30.7 | 24.7 | 27.5 | 9.5 | 7.5 | 9.2 | 33.5 | 7.0 |
| CLIP Score | 51.3 | 41.5 | 30.2 | 25.6 | 26.7 | 8.3 | 7.4 | 9.3 | 33.7 | 6.7 |
| BLIP Score | 61.4 | **49.7** | 36.8 | 28.7 | 29.6 | 9.2 | 8.5 | 9.9 | 37.1 | 7.3 |
| Ours | **64.2** | 49.3 | **37.4** | **28.7** | **29.7** | **11.0** | **12.9** | **11.9** | **44.5** | **8.6** |
| **CC12M (10.4M → 5.20M)** | | | | | | | | | | |
| *Full Size* | 81.0 | 65.0 | 53.5 | 39.2 | 48.8 | 18.1 | 11.4 | 42.7 | 36.1 | 7.6 |
| Random | 74.0 | 59.5 | 48.3 | 34.6 | 43.2 | 19.2 | 11.1 | 43.1 | 35.8 | 7.5 |
| CLIP Score | 77.7 | 62.1 | 49.7 | 36.0 | **51.3** | 21.0 | 13.8 | 49.6 | 43.6 | 8.7 |
| BLIP Score | 79.6 | 62.2 | 53.1 | 37.6 | 48.7 | 19.6 | 16.9 | 59.1 | 50.2 | 9.1 |
| Ours | **81.6** | **64.0** | **53.5** | **38.5** | 48.5 | **21.8** | **18.0** | **59.1** | **54.8** | **10.4** |

Table 1: Zero-shot performance of BLIP models pre-trained using various filtering strategies on Conceptual Captions (CC3M) and Conceptual 12M (CC12M) datasets. Our method outperforms Random, CLIP Score, and BLIP Score across most tasks. Notably, in captioning tasks, our method delivers the best results at both scales, with improvements of 5.0% and 11.1% over the full-sized dataset, respectively.

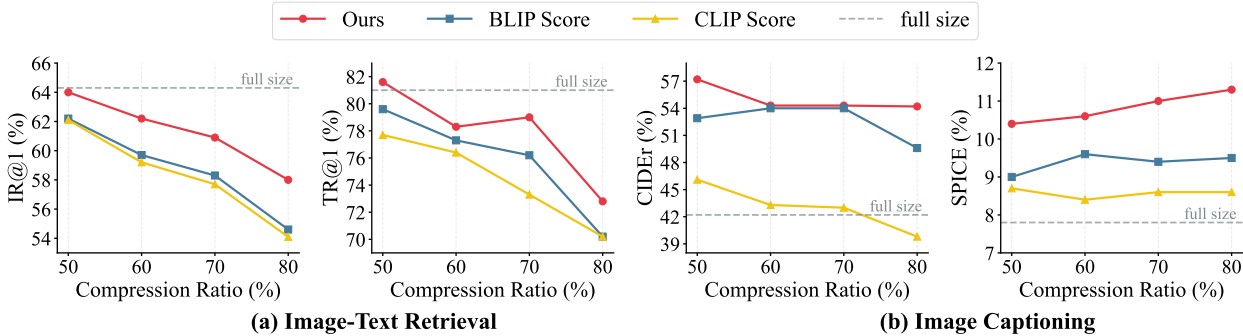

(a) **Image-Text Retrieval**  (b) **Image Captioning**

Figure 4: Performance of different compression ratios filtering by CLIP Score, BLIP Score, and our method. Our method achieves the best performance at all scales. Due to the inherent size of the dataset, an excessively low compression rate can harm performance.

## 4.2 Curating Better Data Samples

We conduct experiments on widely used multimodal datasets, CC3M and CC12M, to demonstrate that our method can curate higher-quality data samples compared to other filtering strategies. Additionally, we investigate the trade-off between data efficiency and vision-language pre-training performance.

**Main results.** Our key results are demonstrated in Table 1. Notably, our method consistently excels on most tasks compared with other filtering strategies and even surpasses the performance achieved using the entire dataset. Specifically, our method outperforms other filtering strategies by an average of at least 1.75% in retrieval tasks, 0.9% in classification tasks, and 1.75% in captioning tasks, highlighting the superiority of incorporating human knowledge in image-text alignment. As the dataset size scales from 3M to 12M, the average performance gains of our method over the full-sized dataset increase from 0.8% to 1.7% in classification tasks and from 5.0% to 11.1% in captioning tasks. These results demonstrate that effectively filtering out noisy samples significantly enhances the performance of vision-language pre-training. Our method preserves visual diversity while refining textual quality during dataset filtering, leading to consistent improvements across various downstream tasks.

| Method | MSCOCO | | Flickr30k | |
|---|---|---|---|---|
| | TR@1 | IR@1 | TR@1 | IR@1 |
| *CC3M+CC12M+LAION-400M (128M → 23M)* | | | | |
| *Full Size* | 54.7 | 41.5 | 78.6 | 66.9 |
| CLIP Score | 54.4 | 41.0 | 75.8 | 67.2 |
| BLIP Score | 53.7 | 40.7 | 80.0 | 67.0 |
| Ours | **56.7** | **42.6** | **82.4** | **69.6** |

Table 2: Zero-shot performance of BLIP models pre-trained on compressed CC3M, CC12M, and LAION-400M datasets. Despite filtering out 83% of the entire dataset, only our method surpasses the performance of models trained on the full-sized dataset, highlighting the advantage of incorporating human knowledge into the data filtering.

| Method | MSCOCO | | | | NoCaps | | | |
|---|---|---|---|---|---|---|---|---|
| | | | | | Valid | | Test | |
| | BLEU@4 | CIDEr | METEOR | SPICE | CIDEr | SPICE | CIDEr | SPICE |
| *CC3M+CC12M+LAION-400M (128M → 15M)* | | | | | | | | |
| *Full Size* | 9.1 | 46.7 | 14.0 | 10.3 | 41.6 | 7.6 | 40.0 | 7.5 |
| CLIP Score | 11.5 | 50.2 | 15.0 | 11.1 | 45.5 | 8.2 | 43.4 | 8.1 |
| BLIP Score | 13.2 | 55.4 | 16.3 | 12.0 | 51.5 | 8.5 | 49.0 | 8.5 |
| Ours | **19.8** | **68.3** | **19.9** | **14.6** | **61.7** | **10.3** | **59.9** | **10.3** |

Table 3: Zero-shot performance of BLIP models pre-trained on sub-datasets containing only 11% of the total samples. Our method not only outperforms other filtering strategies but also exceeds the performance of models trained on the full-sized dataset.

| Dataset | Method | #Samples | Image-Text Retrieval | | | | Image Captioning | | | |
|---|---|---|---|---|---|---|---|---|---|---|
| | | | MSCOCO | | Flickr30k | | MSCOCO | | NoCaps | |
| | | | TR@1 | IR@1 | TR@1 | IR@1 | BLEU@4 | CIDEr | CIDEr | SPICE |
| CC+LAION | *Full Size* | 128M | 54.7 | 41.5 | 78.6 | 66.9 | 9.1 | 46.7 | 40.0 | 7.5 |
| | Ours | 23M | **56.7** | **42.6** | **82.4** | **69.6** | **20.1** | **71.5** | **62.0** | **9.4** |
| | | 15M | 54.1 | 40.7 | 81.9 | 67.7 | **19.8** | **68.3** | **59.9** | **10.3** |

Table 4: Relationship between dataset size on downstream task performance. In retrieval tasks, a decrease in data volume leads to a noticeable drop in performance, highlighting the importance of diversity brought by larger datasets. In contrast, captioning tasks maintain superior performance even with reduced data, indicating that the quality of the data is more critical than its quantity.

**Trade-off between efficiency and performance.** We extend the compression ratio to 80% and report the results in Figure 4. The results demonstrate that our method consistently achieves the best performance at all scales and the performance degrades when only small fractions of the dataset are used. Balancing the efficiency and performance, the optimal performance comes from selecting 50% of the entire dataset and the detailed results are shown in Table 1. Interestingly, in the captioning tasks, the CIDEr remains largely unaffected by the reduction of data samples and SPICE surprisingly increases and even achieves the best when remaining only 20% of the whole dataset. This suggests that the principle of "less is more" applies to certain tasks, warranting further exploration of the trade-off between efficiency and performance. However, given the inherent limitations of the 12M dataset, excessively low compression rates inevitably harm performance. To delve deeper, we turn our attention to the larger-scale dataset including LAION-400M, SBU captions, COCO, and Visual Genome, for further investigation.

| Dataset | Method | #Samples | MSCOCO | | NoCaps | | | |
| --- | --- | --- | --- | --- | --- | --- | --- | --- |
| | | | | | Valid | | Test | |
| | | | CIDEr | SPICE | CIDEr | SPICE | CIDEr | SPICE |
| COCO+VG+CC | Cap-Filt | 129M | 95.2 | 17.1 | 74.4 | 10.7 | 72.2 | 10.5 |
| +SBU+LAION | Ours | 17.6M | **104.4** | **19.0** | **74.8** | **11.1** | **73.3** | **11.1** |

Table 5: Comparion with Captioning-then-Filtering method (Cap-Filt). Our method surpasses Cap-Filt with only 15% of the data Cap-Filt uses, demonstrating the superiority of our method.

### 4.3 Scaling to Large-scale Datasets

This section investigates the data efficiency of large-scale datasets, including CC, LAION-400M, SBU, COCO, and VG. We explore three key questions closely related to vision-language pre-training:

- *How do these datasets perform under extreme compression i.e. removing more than 80% of the entire dataset?*

- *What is the relationship between dataset size and downstream task performance?*

- *Can dataset filtering surpass dataset rewriting?*

**20% beats 100%.** We filter out more than 80% of the entire dataset consisting of CC3M, CC12M, and LAION-400M to achieve a highly compact sub-dataset. The key results are shown in Table 2 and Table 3. Our method achieves an average improvement of 2.85% on retrieval tasks and 11.5% on captioning tasks compared to the full-sized dataset. This is notably higher than the -0.07% and 2.1% improvements with the BLIP Score and the -0.3% and 9.0% improvements with the CLIP Score. Remarkably, our method, using only 20% of the entire dataset, outperformed the results obtained using the full dataset. These findings have significant implications for effectively scaling down multimodal data, enabling more efficient vision-language training.

**Diversity and high-quality matters more.** We investigate the relationship between dataset size and downstream task performance, as presented in Table 4. Previous findings demonstrate that the removal of noisy and misaligned samples significantly enhances the downstream performances. However, this improvement comes with a trade-off. Specifically, in retrieval tasks, reducing the dataset size from 23M to 15M results in a performance decline, with some benchmarks showing worse outcomes compared to the full-sized dataset. Based on the observation, we infer that excessive reduction in the dataset may compromise the diversity of image-text pairs, thereby degrading retrieval task performance. Conversely, in captioning tasks, the reduced dataset consistently outperforms its full-sized counterpart, indicating that the quality of textual captions may play a more critical role in this domain. These findings underscore the delicate balance between dataset size and quality in downstream tasks.

**Filtering beats Captioning-then-Filtering.** Researchers have noticed that the noise in large-scale datasets leads to suboptimal performance (Li et al., 2022; 2023). Therefore, they propose a captioner to generate synthetic captions based on images first, and then, utilize a filter to remove noisy captions from both the original and synthetic captions. Cap-Filt has shown promise in bootstrapping model performance via captioning and filtering. To demonstrate the effectiveness of our method further, we compare our method (*i.e.*, filtering only) with Cap-Filt (*i.e.*, captioning-then-filtering).

As shown in Table 5, our method outperforms the Cap-Filt approach across all metrics, despite using only 15% of the data volume. This striking efficiency can be attributed to the quality of our human-preference-based reward model. Unlike Cap-Filt, which relies on the entire MSCOCO dataset (including potentially noisy web-collected captions) for training its captioning and filtering models, our approach leverages a carefully curated subset of samples aligned with human preferences. This results in a more discerning selection process, enabling our method to identify truly high-quality samples from large-scale datasets.

| Vision Backbone | Method | Image-Text Retrieval | | | | Image Captioning | | | |
|---|---|---|---|---|---|---|---|---|---|
| | | Flickr30K | | MSCOCO | | MSCOCO | | NoCaps | |
| | | TR@1 | IR@1 | TR@1 | IR@1 | BLEU@4 | CIDEr | CIDEr | SPICE |
| ViT-B/16 | *Full Size* | 81.0 | 65.0 | 53.5 | 39.2 | 11.4 | 42.7 | 36.1 | 7.6 |
| | CLIP Score | 77.7 | 62.1 | 49.7 | 36.0 | 13.8 | 49.6 | 43.6 | 8.7 |
| | BLIP Score | 79.6 | 62.2 | 53.1 | 37.6 | 16.9 | 59.1 | 50.2 | 9.1 |
| | Ours | **81.6** | **64.0** | **53.5** | **38.5** | **18.0** | **59.1** | **54.8** | **10.4** |
| ViT-L/16 | *Full Size* | 81.1 | 68.4 | 54.6 | 40.9 | 13.1 | 52.2 | 41.8 | 8.0 |
| | CLIP Score | 79.3 | 64.7 | 52.3 | 38.7 | 15.3 | 55.4 | 46.1 | 8.8 |
| | BLIP Score | 81.5 | 68.1 | 54.8 | 40.8 | 19.0 | 66.6 | 54.8 | 9.5 |
| | Ours | **83.5** | **66.9** | **56.8** | **41.7** | **20.3** | **68.2** | **60.5** | **11.0** |

Table 6: Performance of BLIP models of ViT/B-16 and ViT-L/16 trained on datasets using different filtering strategies. Our method consistently outperforms other filtering strategies and achieves more performance gains compared with full-sized datasets when the vision backbone increases.

| Method | Image-Text Retrieval | | | | Image Classification | | |
|---|---|---|---|---|---|---|---|
| | Flickr30K | | MSCOCO | | ImNet-1K | ImNet-A | ImNet-V2 |
| | TR@1 | IR@1 | TR@1 | IR@1 | Acc. | Acc. | Acc. |
| *CC3M (2.82M → 1.41M)* | | | | | | | |
| *Full Size* | 27.4 | 19.0 | 12.5 | 9.6 | 16.0 | 3.6 | 13.2 |
| Random | 24.2 | 16.8 | 12.4 | 9.3 | 13.7 | 3.5 | 11.2 |
| CLIP Score | 23.4 | 16.6 | 11.9 | 8.9 | 14.1 | 3.0 | 12.1 |
| BLIP Score | 28.9 | 19.6 | 15.2 | 10.2 | **15.6** | 2.9 | 12.9 |
| Ours | **31.0** | **21.0** | **15.9** | **11.3** | 15.0 | **4.1** | **13.0** |

Table 7: Performance of CLIP models of ViT-B/16 trained on different sub-datasets. Our method achieves the best performance on most tasks, demonstrating the generalization of our method to various model architectures.

These findings underscore the potential of our human-centric approach to revolutionize data curation for vision-language tasks, offering a more efficient and effective alternative to existing methods.

## 4.4 Ablation Study

We conduct ablation studies to demonstrate that our method generalizes well to vision-language models of different vision backbones and architectures.

**Vision backbone.** To assess the generalizability of our method across different model architectures, we conduct experiments on BLIP models using the compressed dataset with varying vision backbones. The results are shown in Table 6. Our method consistently outperforms other filtering strategies at both scales. Scaling from ViT-B/16 to ViT-L/16, our method shows an average improvement of 2.8% in retrieval tasks and 4.4% in captioning tasks, compared with 1.5% and 4.2% improvements achieved by the full-sized dataset. It demonstrates our method aligns well with the scaling law of model parameters.

**Model Archtecture.** To further validate the versatility of our method, we extended our experiments to include the CLIP model with a ViT-B/16 architecture. Table 7 demonstrates the experiment results. Our method with the 50% dataset outperforms the full-size dataset on the retrieval task. For instance, we observe an increase in performance from 27.4% to 31.0% in IR@1 and from 19.0% to 21.0% in TR@1. For the classification task, our method demonstrates a good performance on ImageNet-A and ImageNet-V2 with +0.5% increase and slightly −0.2% gap, showing the generalization to different distributions. All results suggest the scalability of our method to different model architectures.

### 4.5 Visualization

**Sub-datasets filtered by different strategies.** As shown in Table 8, Table 9, and Table10, we randomly demonstrate some image-text pairs from the CC3M dataset which rank top 10% according to our method, BLIP-Score, and CLIP-Score. The textual captions of the samples filtered by our method are much longer, containing more visual objects and details. First, the visualization is consistent with the designed aspects for image-text alignment. Second, our method effectively explores complex but high-quality image-text pairs compared with BLIP Score and CLIP Score.

**COCO-HF dataset.** As shown in Table11, we demonstrate the visualization of the proposed COCO-HF dataset.

## 5  Conclusion

In this paper, we delve deep into building datasets with high-quality image-text pairs. We present COCO-HF, an image-text dataset with human knowledge for alignment, and reveal that human knowledge enhances both data efficiency and visual-textual alignment. Scaling up to different datasets, we identify that our method shows significant performance on large-scale and noisy datasets. Scaling to various downstream tasks, we discover that leveraging less but high-quality data leads to a greater abundance of fine-grained knowledge and is more suitable for captioning tasks. These findings underscore the potential of human knowledge in efficiency and alignment. We hope that our work sheds light on the data-centric and human-centric field in the vision-language community and engages more and deeper exploration.

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

| Image | Caption |
|---|---|
|  | a rusted water tank on top of a brick building against a blue sky |
|  | children, happy laughing boy and cute curly little girl having fun at pillow fight with feathers in the air |
|  | person jumps to catch one of many letters floating in the air around his living room as his family looks on in horror |
|  | silhouette of a fallen branch on an old oak tree, sun behind with blue sky and light fluffy clouds, october |
|  | melting butter and dark chocolate together in a pan, stirring with a wooden spoon |
|  | a figure posed with head resting on hand, lost in thought, with a background of several colorful light bulbs symbolizing big ideas he is dreaming up and inventing to solve a need or problem |

Table 8: Visualization of image-text pairs from CC3M dataset which ranks top 10% in our method.

| Image | Caption |
|---|---|
|  | chocolate and blueberry cake with powdered sugar that says relax |
|  | young students working together at the library |
|  | young woman in an apron in her kitchen taking food out of the refrigerator |
|  | a nice christmas greeting card with a red background full of flares and lights. |
|  | businessman with his arms raised in a park |
|  | elderly woman perusing the dahlia 's on display at a flower show |

Table 9: Visualization of image-text pairs from CC3M dataset which ranks top 10% in BLIP Score.

| Image | Caption |
|:---:|:---:|
|  | pregnant chipmunk on a log |
|  | painting of noble person as a girl |
|  | black line art illustration of a person shrugging |
|  | a woman riding a camel on a sand dune |
|  | cats on roof of the building at moon in the night |

Table 10: Visualization of image-text pairs from CC3M dataset which ranks top 10% in CLIP Score.

| Image | Captions |
| --- | --- |
| 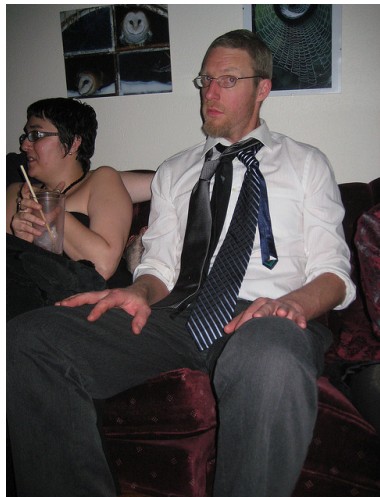 | A man wearing a tie sitting next to a woman wearing glasses sitting on a couch. There is a man sitting on the couch next to a woman but he has three neck ties on. A man and a woman are sitting on a couch together. The man is wearing a tie, and the woman is wearing a dress. There are two other people in the room, one of whom is sitting on the couch next to the couple, while the other person is standing nearby. A man wearing three different neck ties, sitting on a couch. a man and woman sitting on a couch. There are two people sitting on a couch with one holding a drink. A man on a couch who is wearing several ties. There is a man that is sitting on the couch A man wearing 3 ties sitting on a couch. |
| 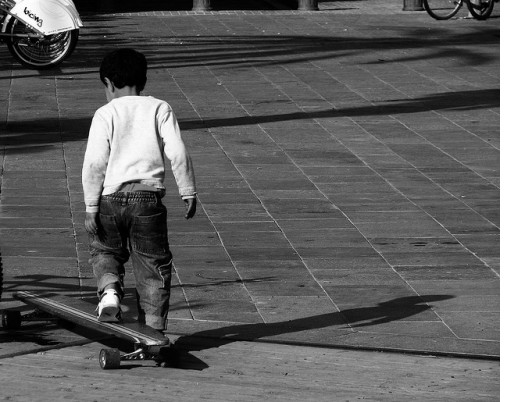 | a little boy stands on a skateboard as a young person walks by A young man riding on top of a skateboard. a young boy is riding a skateboard down a sidewalk A young child skateboarding on a paved walkway. there is a young boy that is riding a skateboard A very small boy playing with a skateboard. boy on skateboard in the middle of a sidewalk with a bike in the background. a boy riding a skateboard on a sidewalk A little boy is standing on a skateboard as he skates along a path. |
| 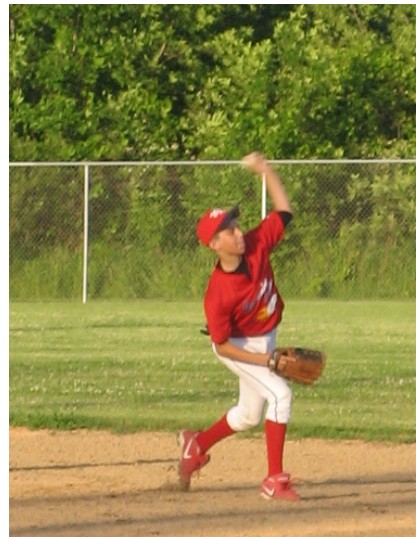 | a young man playing baseball pitching a ball at a baseball game in a field a baseball player throws a ball A young baseball player is in the process of throwing a ball during a game. He is wearing a red and white uniform, and his glove is visible as he prepares to release the ball. There are several other players on the field, some of whom are positioned closer to the pitcher than others. A boy with a glove throwing a baseball on a field. A child pitcher gets ready to throw the baseball The young boy is playing a game of baseball. The young baseball player is throwing a pitch. A baseball player catching a ball on the field boy in red and white baseball uniform throwing a baseball |

Table 11: Visualization of COCO-HF dataset.

