# OpenReview forum: "Filter & Align: Leveraging Human Knowledge to Curate Image-Text Data"
_TMLR — Rejected by TMLR_

### Review · Reviewer_34wQ · 2024-09-22

**Summary Of Contributions:**

The authors propose a filtering technique for image-text pair datasets. By training a reward model on human preferences, the authors filter image-text samples and automatically create a small dataset for fine-tuning. While the dataset has fewer data points w.r.t. baselines, the authors demonstrate it can be used to achieve better performance.

**Audience:**

Yes

**Broader Impact Concerns:**

No concerns about the broader impact of this work.

**Claims And Evidence:**

Yes

**Requested Changes:**

- Add comparison with SemDeDup [1], especially when tuning on LAION. It is well-known that LAION suffers from a duplication problem, but it is unclear if the reward model works in this direction or keeps most duplicates in the filtered set. Also, if possible, the authors should consider adding additional baselines (potentially inspired by Data Filtering Networks [2]). Overall, I believe there is a lack of baselines and the authors should add new ones to confirm the efficacy of their approach.
- Report statistics on concepts pre- and post-filtering. For CLIP-like models, the amount of concepts is a vital component to guarantee good performance on downstream tasks [3]. The reward model does not guarantee any concept balancing and is biased toward object-centric images (i.e., the reward model is trained on a subset of COCO). The authors should report the number of concepts, i.e., nouns, adjectives, and verbs, for the different filtering techniques, to quantify the diversity preserved in the subsets.
- Report additional experiments where the training time increases linearly with the data size. It is well-known that increasing the data pool size without increasing the training time is suboptimal and may be disadvantageous for the largest datasets, i.e., the least filtered. The authors should provide additional results where they train for an increasing number of epochs.


[1] Abbas, Amro, et al. "Semdedup: Data-efficient learning at web-scale through semantic deduplication." arXiv preprint arXiv:2303.09540 (2023). \
[2] Fang, Alex, et al. "Data filtering networks." arXiv preprint arXiv:2309.17425 (2023). \
[3] Udandarao, Vishaal, et al. "No" zero-shot" without exponential data: Pretraining concept frequency determines multimodal model performance." arXiv preprint arXiv:2404.04125 (2024).

**Strengths And Weaknesses:**

Strengths:
- The presentation is curated in both the writing and the visuals.
- While simple, the method is grounded in language modeling literature and demonstrates promising results.
- While the method removes 80% or more of the data size, it retains or improves performance w.r.t. the entire data pool.

Weaknesses:
- Lack of baselines, i.e., only naive filtering approaches are considered as competitors for the proposed method.
- The reward model lacks any concept balancing mechanism, i.e., entire concepts could be removed from a dataset.
- It is unclear why BLIP is the backbone of the reward model.
- The comparison with baseline methods may be unfair, as data size must scale with compute time.

---

> ### Author Response · Authors · 2024-10-16
> **Re: Review of Paper3236 by Reviewer 34wQ**
>
> Thank you for your insightful feedback on our paper. We appreciate the opportunity to clarify and discuss the points raised in your comments.
>
> **Question 1:** *Add comparison with SemDeDup [1], especially when tuning on LAION. It is well-known that LAION suffers from a duplication problem, but it is unclear if the reward model works in this direction or keeps most duplicates in the filtered set. Also, if possible, the authors should consider adding additional baselines (potentially inspired by Data Filtering Networks [2]). Overall, I believe there is a lack of baselines and the authors should add new ones to confirm the efficacy of their approach.*
>
> **Answer 1:** Thanks for bringing in these related works. When comparing to SemDeDup, we would like to highlight that our work takes a different approach—no cross-pair comparisons are conducted in our method; instead, our focus is purely on assessing whether the data itself is of high quality. Therefore, it is highly possible to see a further performance boost when combining ours and SemDeDup.
> Regarding DFN, our methods are not contradictory but rather complementary. DFN still relies on the CLIP score, and our argument is that the CLIP score is not perfect. This suggests that once we have a DFN based on the CLIP score, we could further improve it using our method. We are happy to provide a more detailed discussion of this in our future plans and incorporate it into the next version.
>
> ***
>
> **Question 2:** *Report statistics on concepts pre- and post-filtering. For CLIP-like models, the amount of concepts is a vital component to guarantee good performance on downstream tasks [3]. The reward model does not guarantee any concept balancing and is biased toward object-centric images (i.e., the reward model is trained on a subset of COCO). The authors should report the number of concepts, i.e., nouns, adjectives, and verbs, for the different filtering techniques, to quantify the diversity preserved in the subsets.*
>
> **Answer 2:** We count the number of concepts in the CC3M subsets obtained through different filtering methods. We employ random selecting, CLIP-Score, BLIP-Score, and proposed method to filter out 50% of CC3M. ​​We use NTK to tokenize the captions in the filtered dataset and count the number of verbs, nouns, and adjectives. The results are shown as follows:
>
> |   | **noun** | **adjective** | **verb** |
> |------------|-----------|------------|--------------------------|
> | Random   | 5,337,345 | 1,289,587 | 1,516,616 |
> | CLIP-Score    | 5,240,947 | 1,383,270 | 1,357,098 |
> | BLIP-Score   | 5,508,135 | 1,389,971 | 1,441,879 |
> | Ours    | **5,933,093** | **1,536,032** | **1,641,693** |
>
> We will add these additional results in the next version. The dataset filtered by our method includes the largest number of nouns, adjectives, and verbs, demonstrating the diversity of the selected captions.
>
> ***
>
> **Question 3:** *Report additional experiments where the training time increases linearly with the data size. It is well-known that increasing the data pool size without increasing the training time is suboptimal and may be disadvantageous for the largest datasets, i.e., the least filtered. The authors should provide additional results where they train for an increasing number of epochs.*
>
> **Answer 3:** Thanks for your suggestion. We present the changes in performance as the dataset size and training time increase. We can observe that the performance improves as the training time linearly increases and the data size expands. We will add these additional results in the next version.
>
> | **Dataset** | **#Samples** | **epoch=12** | **epoch=14** | **epoch=16** | **epoch=18** | **epoch=20** |
> |------------|-----------|------------|--------------------------|------------|--------------------------|--------------------------|
> | CC3M | 1.4M | 36.3 | 36.5 | 36.8 | 37.2 | 37.4 |
> | CC12M | 5.2M | 52.9 | 53.2 | 53.2 | 53.5 | 53.6 |
>
> *Table 1: TR@1 on MSCOCO dataset*
>
> | **Dataset** | **#Samples** | **epoch=12** | **epoch=14** | **epoch=16** | **epoch=18** | **epoch=20** |
> |------------|-----------|------------|--------------------------|------------|--------------------------|--------------------------|
> | CC3M | 1.4M | 28.3 | 28.6 | 28.7 | 28.7 | 29.0 |
> | CC12M | 5.2M | 37.8 | 37.9 | 38.2 | 37.9 | 38.5 |
>
> *Table 2: IR@1 on MSCOCO dataset*

---

### Review · Reviewer_ZwKg · 2024-09-23

**Summary Of Contributions:**

- this paper analyses a potential problem existing in current large-scale visual-language training: 1) large-scale visual-language dataset inevitably introduces significant variability in data quality; 2) currently used dataset filtering metrics such as CLIP Score and BLIP Score may be developed upon noisy data, and may reflect machine-based assessments instead of human preference.
- this paper proposes a data filtering strategy considering human judgment through a reward model trained on elaboratly collected small dataset.

**Audience:**

Yes

**Broader Impact Concerns:**

There is no concern on the ethical implications of the work.

**Claims And Evidence:**

No

**Requested Changes:**

- It would be helpful for the authors to clarify how the scores in Figure 1 were computed. The caption states that the proposed method reduces the training sample size from 130M to 15.5M, but it is unclear how the method compares to others, such as CLIP score or models trained with the full dataset.
- In Figure 2, three methods for collecting captions for human annotators are mentioned: web crawling, model generation, and recaptioning. However, the text later states that captions are sourced from MSCOCO sampling, model generation, and human recaptioning. Clarification on the actual caption collection process would be appreciated.

**Strengths And Weaknesses:**

### Strengths:
- The paper introduces a human-annotated image-caption dataset that incorporates human preferences into a reward model, enabling the collection of high-quality data. This effectively reduces the training scale for certain vision-language tasks.
- The proposed method achieves satisfactory performance across multiple vision-language tasks.
- The paper provides extensive experimental analysis, contributing valuable insights into the field.

### Weaknesses:
- As highlighted in the paper, the reward model is trained on a relatively small human-annotated image-caption dataset. This raises concerns about the dataset’s diversity, which could impact the effectiveness of the reward model. Specifically:
  1. If the dataset lacks diversity, the reward model may struggle to learn nuanced human preferences, as the captions may not be sufficiently distinguishable.
  2. The reward model may face challenges when handling instances that were not encountered during training, such as different captioning styles or unseen categories.

   Based on the information provided, the diversity of the human feedback dataset does not appear to be guaranteed. Since mainstream captioning models often use data from MSCOCO, there is a possibility that the generated captions may have a style similar to MSCOCO, and the human recaptioning process is based on the first 2 kinds of captions.

---

> ### Author Response · Authors · 2024-10-16
> **Re: Review of Paper3236 by Reviewer ZwKg**
>
> We are grateful for your thoughtful review and the opportunity to address the concerns raised regarding our paper.
>
> **Question 1:** *It would be helpful for the authors to clarify how the scores in Figure 1 were computed. The caption states that the proposed method reduces the training sample size from 130M to 15.5M, but it is unclear how the method compares to others, such as CLIP score or models trained with the full dataset.*
>
> **Answer 1:** In Figure 1, we compare the performance of BLIP-B/16 trained on the full dataset (referred to as "full-sized" in Figure 1) with that of models trained on datasets filtered using various methods. The filtering methods include CLIP-Score, BLIP-Score, and our proposed method. The datasets used in this comparison are a combination of CC3M, CC12M, and LAION-400M. For the LAION-400M dataset, following prior work[1], we pre-filter images with a resolution lower than 256x256. To ensure a fair comparison, we apply the same training parameters across all filtering methods, including the training epochs, learning rate, and data augmentation strategies, with each model trained to full convergence.
>
> ***
>
> **Question 2:** *In Figure 2, three methods for collecting captions for human annotators are mentioned: web crawling, model generation, and recaptioning. However, the text later states that captions are sourced from MSCOCO sampling, model generation, and human recaptioning. Clarification on the actual caption collection process would be appreciated.*
>
> **Answer 2:** We apologize for any confusion caused by the unclear terminology in the paper. The "web crawling" mentioned in Figure 2 refers to what we later describe as "COCO sampling," since the COCO dataset is collected through web crawling. Similarly, "recaptioning" refers to "human generation." We will ensure consistent terminology in the next version of the paper.
>
> ***
>
> **Question 3:** *Based on the information provided, the diversity of the human feedback dataset does not appear to be guaranteed. Since mainstream captioning models often use data from MSCOCO, there is a possibility that the generated captions may have a style similar to MSCOCO, and the human recaptioning process is based on the first 2 kinds of captions.*
>
> **Answer 3:** In addition to the original COCO dataset, we enhance the diversity of the captions using both model generation and human recaptioning. In both cases, only the images were provided as references (i.e., the original COCO captions were hidden to avoid cross-influence), thereby preserving the diversity of the dataset. For model generation, we employ three different models: BLIP, BLIP-2, and InstructBLIP. For human recaptioning, each human volunteer is shown only the images, without any access to existing captions.
>
> ***
>
> **Reference**
>
> [1] Junnan Li, et al. “BLIP: Bootstrapping Language-Image Pre-training for Unified Vision-Language Understanding and Generation.” ICML 2022.

---

### Review · Reviewer_5ug8 · 2024-09-25

**Summary Of Contributions:**

Paper proposes a new method for dataset compression based on alignment with human provided responses. The method is then shown to have a positive impact on model performance in downstream tasks such as 'information retrieval', 'classification' and 'captioning'.

This method is novel and shown to outperform some of the existing methods in this space on the three main tasks considered in this work ('information retrieval', 'classification' and 'captioning'). Interestingly, high rates of compression (upto 80%) show improvement on some performance metrics, suggesting a win-win scenario in model pretraining.

**Audience:**

Yes

**Broader Impact Concerns:**

In the final version of the manuscript please provide more details about the human subject study conducted and whether it was approved by an IRB. It is also common practice in human-subject studies to report on the demographic composition of the recruited participant pool, the way they were recruited, and how they were compensated for the annotation work.

**Claims And Evidence:**

Yes

**Requested Changes:**

Based on the related work:
- Why does the work not compare the proposed method to CiT (Xu et al 2023a) and TL;DR (Wang et al. 2023)? Since it seems like these are the most relevant methods in the literature, it is important to either compare or provide clear reasoning for why the comparison does not make sense.
- Following the description of prior work, the related work section should describe how and why exactly the new work is different from prior dataset compression methods.

On figure 4:

- Please describe what the y-axis labels mean in the caption of the image. It is useful to provide some explanation for how to interpret different values for each metric considered.
- The figure is confusing because it is not clear whether across all plots lower means worse? It would be good to explain that in the caption.
- While the absolute range of the y-axis across the four plots is very different, it would be helpful to standardize the scale (e.g. one step = 5 percentage points) across the four plots.
- Please add error bars to all plots in Figure 4 and 95% confidence intervals to Table 1.


Missing definitions:
- In the last paragraph on Page 2, the paper refers to the BLIP-B/16 model without introducing what that is. Please try to introduce such terms early on.
- RQ3 in the beginning of Section 4.3 asks about ‘dataset rewriting’ but that term has not been formally defined yet. Please define before using.



Research questions stated in Section 4.3:
- How is RQ1 different from RQ2? Are there different metrics of dataset performance that are not related to downstream performance?
- Is there a reason for setting up the research experiments for larger datasets separately? If yes, it would be useful to motivate that in the beginning of section 4.3, as it is not clear in the current manuscript why is this a separate finding from those in Section 4.2
- It is not clear from the manuscript whether the third question is answered to sufficient detail.



Under-supported claims:
- Claim about visual diversity: on page 7 last para it says “Our method preserves visual diversity”. Please provide the evidence to support this.
- In page 9, from paragraph ‘Diversity and high quality matters more’: “We infer that excessive reduction in the dataset may compromise the diversity of image-text pairs..” There exist metrics for quantifying visual or textual diversity in the literature, which would be useful here to support the claims being made. In general, since there is a running conversation in the paper about the diversity of the compressed dataset, it seems important to report some numbers associated with this for all the experiments.
- On page 7: “Notably, our method consistently excels on most tasks compared with other filtering strategies a” → tasks considered in this paper/ in our analysis
- On page 7: “Leading to consistent improvement across downstream tasks” → the improvement is not consistent across all tasks, and it is not clear which method is your method being compared against here: full-size dataset or BLIP-score-based 50% size dataset?
- Caption of Table 2: “Despite filtering out 83% of the entire dataset, only our method surpasses the performance of models trained on the full-sized dataset”. Pls specify that this is on the choice of tasks chosen in this paper.



Missing limitations section:
Important to reflect on the potential limitations of the study and experiment design in this work and its effects on the findings.
Are there any potential downsides of compressing the dataset that this analysis is missing? It paints a win-win picture which seems promising but could be incomplete. It would be useful to understand the authors’ thoughts on this and have it be a part of the limitations section.

**Strengths And Weaknesses:**

Strengths:
1. The work is novel and has a high scope of impact in the space of image-vision pretraining which is a very relevant space of research.
2. The method suggests a high scope of improvement in the current state of the art and is an important work to be introduced to the research area as it is the first to centers human knowledge in the curation of datasets, which does not seem to have been done before. I would invite the authors to correct this statement, if I am wrong.
3. Extensive experiments showing performance outcomes on a variety of downstream tasks, to paint a fuller picture.

I will point out here three main weaknesses and the rest are covered in the requested changes question.

1. **Human Data Analysis**:

The paper mentions the collection of human preference annotations but lacks a detailed analysis of this data. It would be beneficial to include:
 - Analysis of disagreements: Where did annotators disagree most frequently? Analyzing these instances can reveal how human perception of image-text alignment is different from the status quo.
 - The human annotation form asks them to compare the captions on four axes: completeness, accuracy, vividness and context. How are these 4 different measures being aggregated to get a final ranking of the image-caption pairs? These details are very important for a clearer understanding of the method
- Correlation with existing metrics: How well do human preferences correlate with existing metrics like CLIP Score and BLIP Score? This analysis can help understand the limitations of these automated metrics and highlight the value of incorporating human knowledge.

2. **Sub-Dataset Characterization**:

The paper claims to produce a "compact and high-quality" dataset but doesn't provide much insight into the characteristics of this filtered dataset. For a complete writeup the paper should consider including:

- Qualitative examples: Show examples of image-text pairs that were included and excluded by the filtering process. This can help illustrate the practical impact of the proposed method.
- Analysis of distribution shifts: How does the distribution of concepts, styles, or other relevant attributes change after filtering?Understanding these shifts can help assess potential improvements or biases introduced by the filtering process and anticipate its impact on downstream tasks.
- Under the methodology proposed in this paper, it is not clear whether the resulting compact dataset is deterministic or not (which is important to understand). Elaboration on the methodology would help clarify this aspect.

3. **Rigor in deriving claims and results**:

- Statistical significance testing: Include statistical significance tests to support claims of performance improvement. This will help determine whether the observed differences are likely due to the proposed method or random variation.
- Effect size reporting: Report effect sizes (e.g., Cohen's d) to quantify the magnitude of the observed improvements. This will help assess the practical significance of the findings.
- There are several instances where the text overclaims the result, which should be avoided.
For plots and tables, it is important to report error bars and 95% confidence intervals to help the reader understand the significance of the effect, when not reporting p-values.

---

> ### Author Response · Authors · 2024-10-16
> **Re: Review of Paper3236 by Reviewer 5ug8**
>
> We extend our gratitude to Reviewer 5ug8 for your comprehensive review and insightful feedback on our work. Your constructive criticism is highly valued and will undoubtedly aid in refining our research. In response to each of your observations, we have detailed the modifications and enhancements we plan to implement in our work.
>
> **Question 1:** *Based on the related work:  (a) Why does the work not compare the proposed method to CiT (Xu et al 2023a) and TL;DR (Wang et al. 2023)? Since it seems like these are the most relevant methods in the literature, it is important to either compare or provide clear reasoning for why the comparison does not make sense. (b) Following the description of prior work, the related work section should describe how and why exactly the new work is different from prior dataset compression methods.*
>
> **Answer 1:**
>
> Thanks for bringing these two related works. We would like to highlight that these two works are actually complementary to ours.
>
> CiT dynamically curates the data during training a CLIP model on the fly. Since the proposed dynamic compress method is tied to the adjusting distribution of the CLIP model, retraining is necessary for each new dataset. The retraining is resource-intensive and makes the reproduction difficult. In contrast, our method is more practical and efficient, as we only require inference using the pre-trained reward model.
>
> TL;DR initially employs the BLIP model to screen out low-quality image-text pairs coarsely. Subsequently, it learns a codebook to cluster and refine the filtering process. However, storing this trained codebook consumes significant memory, making it impractical for large-scale datasets. Moreover, the proposed method is not an end-to-end solution.
>
> Both CiT and TL;DR are built upon noisy datasets, therefore potentially less effective in filtering data. Given our method’s superior performance in ensuring alignment, we posit that integrating our method with existing frameworks like CiT and TL;DR could significantly bolster data efficiency and performance. We will add these discussions in the next version.
>
> ***
>
> **Question 2:** *On figure 4: (a) Please describe what the y-axis labels mean in the caption of the image. It is useful to provide some explanation for how to interpret different values for each metric considered. (b) The figure is confusing because it is not clear whether across all plots lower means worse? It would be good to explain that in the caption. (c) While the absolute range of the y-axis across the four plots is very different, it would be helpful to standardize the scale (e.g. one step = 5 percentage points) across the four plots.*
>
> **Answer 2:**
>
> In retrieval tasks, IR@1 (Image Retrieval at rank 1) and TR@1 (Text Retrieval at rank 1) are metrics used to evaluate the effectiveness of a model in matching images to corresponding textual descriptions and vice versa. Take the IR@1 as an example. IR@1 measures the accuracy of image retrieval when using text queries. Specifically, it indicates the percentage of times the correct image is retrieved as the top result (i.e., ranked first) for a given text query. A higher score indicates a more effective retrieval model, capable of accurately matching images and text.
>
> In image captioning tasks, CIDEr and SPICE are evaluation metrics that assess the quality of the generated captions by comparing them with human-annotated reference captions. CIDEr measures how well the generated caption aligns with a set of human-annotated reference captions, focusing on the consensus of commonly occurring n-grams (sequences of words) between the generated and reference captions. It is designed to give higher scores when a caption shares more frequent and relevant phrases with human captions. SPICE focuses on the semantic content of the generated captions, specifically how well the caption describes the objects, attributes, and relationships in the image. Unlike CIDEr, which focuses on n-gram overlap, SPICE evaluates the meaning and structure of the caption by comparing it to human captions at a more abstract level.
>
> In Figure 4, we aim to demonstrate that our method performs well across different data proportions. The experimental results show that our method consistently achieves strong performance across various downstream tasks.
>
> The y-axis of each sub-figure represents different metrics, and the absolute ranges vary to ensure better visualization.
>
> We will revise the submission accordingly to make these concepts clearer.

---

> > ### Author Response · Authors · 2024-10-16
> >
> > **Question 3:** *Missing definitions: (a) In the last paragraph on Page 2, the paper refers to the BLIP-B/16 model without introducing what that is. Please try to introduce such terms early on. (b) RQ3 in the beginning of Section 4.3 asks about ‘dataset rewriting’ but that term has not been formally defined yet. Please define before using.*
> >
> > **Answer 3:**
> >
> > Thanks for raising these concerns.
> >
> > To clarify, BLIP-B/16 is a variant of the BLIP (Bootstrapping Language-Image Pre-training) model. BLIP models use a combination of transformers and vision-language pretraining strategies to effectively bridge the gap between textual and visual data. BLIP models come in various sizes, and the main difference between these models lies in the vision encoder they use. BLIP-B/16 uses a ViT-B/16 as its image encoder.
> >
> > "Dataset rewriting" refers to the process of rewriting captions in the Cap-Filt method. Different from our approach, Cap-Filt rewrites the captions in the dataset before applying any filtering.
> >
> > We will revise the submission accordingly to clearly define them.
> >
> > ***
> >
> > **Question 4:** *Research questions stated in Section 4.3: (a) How is RQ1 different from RQ2? Are there different metrics of dataset performance that are not related to downstream performance? (b) Is there a reason for setting up the research experiments for larger datasets separately? If yes, it would be useful to motivate that in the beginning of section 4.3, as it is not clear in the current manuscript why is this a separate finding from those in Section 4.2. (c) It is not clear from the manuscript whether the third question is answered to sufficient detail.*
> >
> > **Answer 4:**
> >
> > RQ1 analyzes the performance of different data filtering methods as the dataset size progressively decreases. Our method performs well even with a small amount of data, indicating its ability to select representative samples from the dataset effectively.
> >
> > RQ2 is analyzed by observing the performance of different downstream tasks as the dataset size varies. When the dataset progressively reduces, the performance of retrieval tasks significantly declines compared to using the full dataset. However, captioning tasks still maintain relatively strong performance. This suggests that retrieval tasks require a certain level of data diversity, while captioning tasks benefit more from smaller but higher-quality data.
> >
> > We separate these two sections to more clearly demonstrate the effectiveness of our method on large-scale datasets. In future versions, we will merge the two sections to make the reading flow more seamlessly.
> >
> > In RQ3, we compared the filtering-only approach (our method) with the caption-then-filter approach (Cap-Filt). Our method filters the original dataset, while Cap-Filt first rewrites the captions in the original dataset and then applies filtering. In other words, Cap-Filt performs filtering on a dataset with improved caption quality. Moreover, Cap-Filt retained 129M data points after filtering, whereas our method retained only 17.6M. Despite the significantly smaller dataset, our method still outperforms Cap-Filt as shown in Table 5. This indicates that existing large-scale image-text datasets contain a sufficient amount of data, but the quality is inconsistent. Identifying high-quality data within these datasets can yield significant benefits.
> >
> > ***
> >
> > **Question 5:** *Claim about visual diversity: on page 7 last para it says “Our method preserves visual diversity”. Please provide the evidence to support this.*
> >
> > **Answer 5:** In Table 1, our method consistently outperforms almost all other filtering methods in image classification and, in some cases, even surpasses the full dataset. Since image classification relies on understanding image patches, it demonstrates that our method retains a diverse set of visual features during the filtering process.
> >
> > ***
> >
> > **Question 6:** *In page 9, from paragraph ‘Diversity and high quality matters more’: “We infer that excessive reduction in the dataset may compromise the diversity of image-text pairs..” There exist metrics for quantifying visual or textual diversity in the literature, which would be useful here to support the claims being made. In general, since there is a running conversation in the paper about the diversity of the compressed dataset, it seems important to report some numbers associated with this for all the experiments.*
> >
> > **Answer 6:** As shown in Table 4, when the dataset size decreases from 123M to 23M, the performance of the retrieval task improves. However, when further reduced from 23M to 15.5M, performance declines. The initial performance gain can be attributed to the removal of low-quality data from the dataset. The subsequent performance drop is likely due to an excessive reduction in dataset size, which impacts data diversity. It is reasonable to assume that continually decreasing the dataset size would eventually affect the performance of downstream tasks.

---

> > > ### Author Response · Authors · 2024-10-16
> > >
> > > **Question 7:** *(a) On page 7: “Leading to consistent improvement across downstream tasks” → the improvement is not consistent across all tasks, and it is not clear which method is your method being compared against here: full-size dataset or BLIP-score-based 50% size dataset? (b) Caption of Table 2: “Despite filtering out 83% of the entire dataset, only our method surpasses the performance of models trained on the full-sized dataset”. Pls specify that this is on the choice of tasks chosen in this paper.*
> > >
> > > **Answer 7:** In Table 1, we conduct evaluation on downstream tasks including image-text retrieval, image classification, and image captioning. Compared to other data filtering methods, our method consistently outperforms them (as highlighted in bold in Table 1.). Notably, in the image classification and captioning tasks, our method even surpasses the performance of the full dataset.
> > >
> > > ***
> > >
> > > **Question 8:** *Missing limitations section: Important to reflect on the potential limitations of the study and experiment design in this work and its effects on the findings. Are there any potential downsides of compressing the dataset that this analysis is missing? It paints a win-win picture which seems promising but could be incomplete. It would be useful to understand the authors’ thoughts on this and have it be a part of the limitations section.*
> > >
> > > **Answer 8:** Our work primarily focuses on compressing pre-training data for training multimodal models. Future work could explore compressing data for instruction tuning. This would require careful attention to the data proportions across different tuning tasks, as well as distribution shift between different domains.

---

### Decision · Action_Editor_sZfC · 2024-11-16

**Recommendation:** Reject

**Comment:**

The paper received mixed reviews from three reviewers. The reviewers generally deemed the proposed method effective for the task setting of this work, and noted that the experimental evaluation is extensive. Specifically, the method achieves high data compression rates on several large image-text datasets and shows mostly positive impacts on model performance across three downstream tasks: Image-Text/Text-Image retrieval, Image Classification, and Captioning. It outperforms several VLM score-based filtering baselines.
However, the reviewers raised several major concerns:
1. Insufficient comparisons to related work (5ug8, 34wQ).
2. Missing analysis on the properties of the filtered dataset (5ug8, 34wQ).
3. Lack of clarity in presentation (5ug8, ZwKg, 34wQ).
4. Overclaims on the efficacy of the method (5ug8).
5. Potential unfair comparisons with the baselines (34wQ).

The authors’ response addressed some of these concerns, particularly 1 and 3. During the discussion phase, two reviewers felt that their major concerns were mostly addressed and leaned towards acceptance in their recommendations. However, one expert reviewer remained unconvinced about concerns 2 and 4, and leaned towards rejection.
Given these borderline recommendations, the AE read the paper, reviews, and discussion, and found that several major concerns raised by the reviewers were not fully addressed. The effectiveness of the proposed strategy remains unconvincing in the following three aspects:
- Insufficient support for the quality of filtered datasets:
The paper lacks a detailed analysis of the filtered datasets, leaving important properties regarding the distribution of concepts and other attributes, such as diversity, unclear. The authors’ response only provided limited statistics, casting doubts on the claimed high quality of the datasets. While the authors argued that the quality could be demonstrated through their experiments, their setting is limited to a specific model architecture (BLIP) and only three types of downstream tasks, raising concerns about its generalization.
- Unclear training settings in experimental comparisons:
The experiments do not provide detailed hyperparameter settings for each method, such as the number of training epochs. As pointed out by Reviewer 34wQ, insufficient training can impact baseline performance, leading to potentially unfair comparisons. The authors’ response did not include a comparison of their method and the baselines with an increasing number of epochs.
- Lack of evaluations on important downstream tasks:
The experiments focus solely on three types of downstream tasks—image/text retrieval, image classification, and captioning—which limits the scope of the claims. Multimodal Language Models (MLLMs) are typically evaluated on various Visual Question Answering (VQA) tasks, as seen in works like BLIP-2 (Li et al., 2023) and LLAVA (Liu et al., 2023), to assess performance on complex reasoning tasks.

Due to these issues, the paper requires a major revision and is not yet ready for publication in TMLR. The authors are encouraged to revise the paper accordingly and resubmit later.

**Audience:**

The paper addresses the efficient learning of large vision-language models, which is an active research topic in machine learning and computer vision. Its findings would be interesting to TMLR's audience in the related areas.

**Claims And Evidence:**

The paper proposes a strategy for image-text dataset compression, leveraging human preferences to curate large-scale noisy datasets into a more compact format. The main claim of this work revolves around the effectiveness of the compression, emphasizing the quality of the resulting datasets, the high compression rate, and the beneficial effects on the performance of finetuned models in downstream tasks.

While the paper presents compelling experimental results demonstrating strong compression rates and improvements in certain downstream tasks, it falls short in providing convincing evidence in three critical areas: 1) insufficient support for the quality of the filtered datasets, 2) unclear training settings in experimental comparisons, and 3) lacking evaluations on important downstream tasks. See the comments below for the details.

**Resubmission Of Major Revision:**

The authors may consider submitting a major revision at a later time.